# Vitamin A Transporters in Visual Function: A Mini Review on Membrane Receptors for Dietary Vitamin A Uptake, Storage, and Transport to the Eye

**DOI:** 10.3390/nu13113987

**Published:** 2021-11-09

**Authors:** Nicasio Martin Ask, Matthias Leung, Rakesh Radhakrishnan, Glenn P. Lobo

**Affiliations:** Department of Ophthalmology and Visual Neurosciences, University of Minnesota, Minneapolis, MN 55455, USA; mart4460@umn.edu (N.M.A.); leung132@umn.edu (M.L.); rakeshr@umn.edu (R.R.)

**Keywords:** vitamin A transporters, all-*trans* retinol, retinyl esters, LRAT, STRA6, RBPR2, RBP4, retinol-binding proteins, photoreceptors, visual function

## Abstract

Vitamins are essential compounds obtained through diet that are necessary for normal development and function in an organism. One of the most important vitamins for human physiology is vitamin A, a group of retinoid compounds and carotenoids, which generally function as a mediator for cell growth, differentiation, immunity, and embryonic development, as well as serving as a key component in the phototransduction cycle in the vertebrate retina. For humans, vitamin A is obtained through the diet, where provitamin A carotenoids such as β-carotene from plants or preformed vitamin A such as retinyl esters from animal sources are absorbed into the body via the small intestine and converted into all-*trans* retinol within the intestinal enterocytes. Specifically, once absorbed, carotenoids are cleaved by carotenoid cleavage oxygenases (CCOs), such as Beta-carotene 15,15’-monooxygenase (BCO1), to produce all-*trans* retinal that subsequently gets converted into all-*trans* retinol. CRBP2 bound retinol is then converted into retinyl esters (REs) by the enzyme lecithin retinol acyltransferase (LRAT) in the endoplasmic reticulum, which is then packaged into chylomicrons and sent into the bloodstream for storage in hepatic stellate cells in the liver or for functional use in peripheral tissues such as the retina. All-trans retinol also travels through the bloodstream bound to retinol binding protein 4 (RBP4), where it enters cells with the assistance of the transmembrane transporters, stimulated by retinoic acid 6 (STRA6) in peripheral tissues or retinol binding protein 4 receptor 2 (RBPR2) in systemic tissues (e.g., in the retina and the liver, respectively). Much is known about the intake, metabolism, storage, and function of vitamin A compounds, especially with regard to its impact on eye development and visual function in the retinoid cycle. However, there is much to learn about the role of vitamin A as a transcription factor in development and cell growth, as well as how peripheral cells signal hepatocytes to secrete all-*trans* retinol into the blood for peripheral cell use. This article aims to review literature regarding the major known pathways of vitamin A intake from dietary sources into hepatocytes, vitamin A excretion by hepatocytes, as well as vitamin A usage within the retinoid cycle in the RPE and retina to provide insight on future directions of novel membrane transporters for vitamin A in retinal cell physiology and visual function.

## 1. Mechanisms Involving Intestinal Absorption of Provitamin A Carotenoids and Preformed Vitamin A

Before the important roles of vitamin A transporters can be discussed, the general schematic of macroscale vitamin A interconversions and movements within the cell and circulation must first be established. The base isoprenoid structure of vitamin A in all forms found in biological processes are derived from two dietary sources: β-carotene from plant sources and retinyl palmitate from animal sources [1,2,3]. Besides the method of intake for these two vitamin A sources, the subsequent pathways after absorption converge. For β-carotene, this macromolecule enters intestine epithelial cells through the scavenger receptor class B, type 1 (SCAR-B1/SR-B1) transporter, where intracellular β-carotene is then cleaved into two molecules of all-*trans* retinal by β-carotene monooxygenase 1 (BCO1/BCMO1), which is then reduced into all-*trans* retinol in the cytosol [4]. For extracellular retinyl palmitate, the molecule is first hydrolyzed by extracellular retinyl esterases into all-*trans* retinol, the alcohol then diffuses into the epithelial cell. The all-*trans* retinol from both sources binds to cellular retinol binding protein 2 (CRBP2). The retinol-CRBP2 complex then interacts with lecithin retinol acyltransferase (LRAT) and converts the retinol into retinyl esters and dissociates with CRBP2 [4,5,6]. These retinyl esters are then packaged into nascent chylomicrons along with other lipids and excreted into general circulation through exocytosis. It should be mentioned that most retinyl esters stay with the nascent chylomicron throughout its process in converting into a chylomicron remnant [7]. Around 70% of these chylomicrons are absorbed into the liver for storage. The remaining chylomicron remnants are absorbed into peripheral organs, thus acting as another pathway for vitamin A delivery in extrahepatic tissues [7,8].

The tissue storage of vitamin A is facilitated by hepatocyte-associated hepatic stellate cells. These two cell types function together and can respond to stimuli indicating low vitamin A in body systems; as such, the hepatocyte-associated hepatic stellate cells are capable of both in-taking vitamin A for storage and releasing vitamin A into lymphatic capillaries and blood vessels. It should be mentioned that the mechanism of function for detection of low concentrations of vitamin A in the peripheral organs is unknown and individual peripheral organs such as the eye are hypothesized to be able to independently signal hepatocytes for vitamin A release [1,2,3]. The arrived retinyl esters are first hydrolyzed by intracellular retinyl ester hydrolases into cellular retinol-binding protein 1 (CRBP1) bound all-*trans* retinol, which then enters the hepatic stellate cell and transformed into storage capable retinyl esters by LRAT. Excretion of vitamin A first involves hydrolysis of storage retinyl esters into all-*trans* retinol, which is then excreted through the hypothesized efflux capabilities of the retinol-binding protein 4 receptor 2 (RBPR2) in the intestines. Ejected all-*trans* retinol then associates with RBP4 and transthyretin to form a complex that allows movement through lymphatic and cardiovascular vessels [1,2,3] (Figure 1).

## 2. Uptake of Carotenoids—SR-B1

SCARB1 or SR-B1, is a 509 amino acid integral membrane protein that facilitates the uptake of many different macromolecules into epithelial cells. Through nuclear magnetic resonance microscopy (NMR), it was found that a leucine zipper dimerization motif found in the trans-membrane domain C-terminal was integral to its ability to bind lipoproteins [9]. As such, SCARB1 is an important regulator of cholesterol metabolism and lipid metabolism, functioning as a receptor for low density, very low density, and high-density lipoproteins [10]. Additionally, SCARB1 can also serve as a transporter for vitamins, including tocopherols, and carotenoids such as β-carotene and xanthophylls [10,11].

The importance of SCARB1 in carotenoid transport was demonstrated through the seminal work from the von Lintig Lab. Fruit flies containing a nonsense mutation in neither inactivation nor afterpotential D (*ninaD*) gene eliminates the expression of the fruit fly SCARB1 analog. These mutant flies displayed significantly lower carotenoid composition in the carotenoid heavy areas of the trunk and head, as well the presence of immature rhodopsin in the retina. Furthermore, a diet supplemented with preformed vitamin A or significantly high amounts of β-carotene was shown to be able to allow for rhodopsin maturation in *ninaD* flies, with both diets bypassing the lack of functional SCARB1 [12].

### 2.1. Carotenoid Cleaving Enzymes—BCO1, BCO2

BCO1 and BCO2 belongs to an enzyme family called carotenoid cleavage oxygenases (CCOs). CCOs are characterized by their ability to cleave the carotenoid polyene backbone with high stereoselectivity and regioselectivity, thus cleaving only selected polyenes at specific sites leaving specific products with very high fidelity [13,14,15]. Due to the hydrophobic nature of its substrates and its storage within hydrophobic liposomes, CCOs contain external regions of α-helices with hydrophobic residues that allow for its interaction with phospholipid bilayers and carotenoid substrates. Another structural characteristic of note is the presence of hydrophobic “tunnels” that allow for the entrance of the hydrophobic carotenoid into the catalytic core [16].

All enzymes in the CCO enzyme family contain a Fe^2+^ in the catalytic center fixed by four highly conserved His residues, which are in turn fixed by three highly conserved Glu residues. The iron catalytic core then activates oxygen for one of two theorized mechanisms of polyene cleavage. The monooxygenase reaction hypothesizes a two-step reaction through an epoxide intermediate and a trans-diol intermediate, while the dioxygenase reaction involves a one-step reaction through a highly unstable dioxetane intermediate [16].

In the β-carotene metabolic pathway, BCO1 catalyzes the cleavage of β-carotene taken in through SCARB1 into two molecules of retinal through symmetrical cleavage of β-carotene. This retinal will then serve as the precursor for all carotenoids found in biological processes, such as retinyl esters for storage in the liver, all-*trans* retinoic acid for use as a ligand for RAR-RXR transcription factors, or 11-*cis* retinol for entry into the visual cycle [13].

While BCO1 functions to cleave β-carotene symmetrically for conversion into biologically useful forms, the information known for BCO2 is comparatively much less. Catalytically, BCO2 is different from BCO1 in that it cleaves β-carotene asymmetrically, creating what is known as apocarotenals. BCO2 also seems to preferentially bind xanthophylls over carotenes. These apocarotenals serve a multitude of functions that are just now being elucidated by various laboratories. Some of these functions include regulation in mitochondrial apoptosis and working in conjunction with BCO1 to generate retinoids from asymmetric β-cryptoxanthin [17].

### 2.2. An Intestinal Transcription Factor Regulates Vitamin A Absorption—ISX

The pathway for absorption of carotenoids is negatively regulated by the homeodomain transcription factor (ISX). The expression of ISX is directly stimulated by the retinoic acid receptor (RAR) and the downstream retinoic acid metabolite, which leads to the downregulation of SCARB1 and BCO1. The repression of these two fundamental proteins in the retinoid pathway by retinoic acid-induced ISX constitutes a negative feedback loop that reduces provitamin A intake while retinoid is plentiful [18].

## 3. Transport of All-*Trans* Retinol to Extrahepatic Organs—RBP4 and Retinol

All-*trans* retinol is the fundamental transport form of vitamin A; additionally, all functional retinoids and vitamin A metabolites are derived from retinol. Alongside retinyl esters, retinol is the most abundant form of vitamin A. Retinyl esters serves as the main storage form of retinol, acts as the primary transport form of newly arrived vitamin A, and acts as the precursor for vitamin A metabolites [19]. The transport of hepatic retinol within the serum is facilitated through its binding to retinol-binding protein 4 (RBP4). RBP4, the transport protein responsible for hepatic retinol transport, is primarily expressed in hepatic liver tissue, where it forms a holo-enzyme complex with the retinol substrate and transthyretin (TTR) that gets mobilized out of the hepatocyte and into circulation. The majority (about 85%) of circulating RBP4 is in a holoenzyme complex with retinol, with the remaining 15% in an apo-RBP4 state [20]. Adipose tissue can express RBP4, but they are not a major source of the holo-enzyme circulating in the body, such as those produced in liver hepatocytes. Mice that have liver RBP4 knocked out displayed a significant decrease in serum RBP4 while maintaining normal adipose production of the transport protein [21]. The complex formed between holo-RBP4 and TTR is shown to enhance the secretion of RBP4 from the liver into circulation, as well as to help stabilize the complex and reduce its likelihood of renal filtration out of the body [22]. However, secretion and tissue concentration of retinoids is not limited by TTR binding of RBP4 or the lack thereof, suggesting alternate methods of retinoid transport (such as lipoprotein transport of retinyl esters or reduced retinoic catabolism) are performed in the lack of sufficient serum RBP4 due to TTR deficiency [22]. 

The concentration of RBP4/Retinol lies in a very narrow and regulated range of around 2–3 uM in humans and 1 uM in mice [23], where fluctuations from this concentration range can lead to dysfunction and disorder. Elevated levels of holo-RBP4 in circulation are correlated to an increase in insulin tolerance, subsequent type 2 diabetes, and obesity, as well as the development of nonalcoholic fatty liver disease (NAFLD), among other metabolic diseases based on several case studies involving observational studies of human patients with RBP4 and retinol deficiencies in circulation, and molecular studies using mouse and cell culture models [20]. However, these findings are controversial, with some counter studies suggesting that there is no link between increased circulating RBP4 and insulin resistance, type 2 diabetes, and NAFLD [20,24]. Decreased concentrations of circulating RBP4 have been linked to night-blindness, which was noted in a couple of case studies where individuals experienced impaired vision and retinal dystrophy from low RBP4 serum concentration, which was likely due to mutations found in RBP4 or the complete lack of the protein. Furthermore, reduced RBP4 serum concentration does not appear to lead to any abnormal phenotype other than impaired visual function [24]. In the absence of TTR, thus leading to a decrease in serum holo-RBP4 concentration comparable to vitamin A deficiency, vision impairment was not observed in TTR deficient mice, and the mice were phenotypically normal all around. This lack of phenotype is likely due to the increased ability of RBP4 to bind to the retinol transporter (which has a high affinity for the RBP) in the absence of TTR, compensating for the transport proteins decreased serum concentration with the increased influx into retinal tissue [25]. In the complete absence of RBP4, from the work of Quadro and colleagues in 1999, it was found that normal visual development and function can be sustained with sufficient vitamin A intake in mice from alternate pathways of retinoid transport, but disease phenotypes were observed in RBP4 null mice in times of vitamin A deficiency likely due to the lack of hepatic mobilization of retinoids in storage [26]. The impact of RBP4 concentration in circulation and vision is more directly related to the transport protein itself and its functions rather than the other proteins that interact with RBP4.

## 4. Known Vitamin A Transporters/RBP4 Receptors

### 4.1. Stimulated by Retinoic Acid 6 (STRA6)

Transport of retinoids in and out of cells has been hypothesized to be facilitated by cell surface receptor transport proteins since the initial report of a cell surface receptor for RBP in the 1970s [27]. Despite their initial reports in the 1970s, the cell surface receptor for RBP bound retinol was not characterized for another three decades until the Sun lab in UCLA identified stimulated by retinoic acid 6 (STRA6) in the retinal pigment epithelium (RPE) as a major cell surface transporter for holo-RBP/Retinol in 2007 [28]. The gene for STRA6 was originally identified in 1995 with an unknown function [29]. STRA6 was later postulated to be an integral membrane protein and was found to be highly expressed in rodent embryos and, in rodent adults, the eye (RPE), brain, kidneys, spleen, testes, and female genital tract [30].

The proper function and characterization of STRA6 as the major RBP-bound retinoid transport protein was elucidated a decade later with the work of Kawaguchi et al. in 2007, where they found the membrane protein to regulate the uptake of RBP-bound retinol in bovine RPE cells (Figure 1). STRA6 binds to the holo-RBP complex with high affinity (K_d_ = 59 nM) and shuttles the RBP-bound retinol across the membrane [28]. STRA6 transports retinol bidirectionally depending on RBP’s and LRAT concentrations intracellularly and holo/apo-RBP’s extracellularly, with influx occurring in the presence of LRAT, apo-CRBP1, and holo-RBP, and efflux in the presence of holo-CRBP1 and apo-RBP [31]. Much of these previous findings were made without knowledge of the structure of STRA6, which was believed to be novel for an integral protein until 2016. The solved structure of STRA6 through cryo-EM reinforced some of the findings of the function of STRA6, as well as providing new insights and questions about the regulation with CaM and other functional sites of STRA6. Analysis of the structure finds that RBP binds to an extracellular site on STRA6 and shuttles the retinol molecule into the lipid bilayer, where passive diffusion into the cell likely occurs along with subsequent uptake by CRBP1 [32]. 

Mutations in STRA6 can lead to a myriad of diseases and phenotypes. STRA6 mutations during development can lead to anophthalmia, microphthalmia, and other symptoms that overlap with phenotypes associated with Matthew-Wood syndrome [33]. Matthew-Wood syndrome is a rare congenital disease associated with microphthalmia, mental deficiencies, and various organ deformities [34]. However, these phenotypes can range from mild to severe depending on the mutations of STRA6 and are not explicitly caused by STRA6 mutations but by an excess of retinoic acids in some cases [34]. The associations between mutations in STRA6 and incidence of Matthew-Wood syndrome have been used to create animal models for Matthew-Wood Syndrome in later research, such as with the work of the von Lintig lab in 2008, where they found that excess holo-RBP4 in circulation disrupts vitamin A uptake and causes developmental abnormalities, such as those seen in Matthew-Wood Syndrome, using a morpholino approach to generate a Stra6-deficient zebrafish model [35]. Further research on STRA6 mutations was conducted using mammalian models with STRA6 knockout mice. In 2012, the Bok lab in UCLA generated a STRA6 knockout mouse model to find changes in their visual function and development, finding that the mutant mice had phenotypic differences from controls such as reduced rod and cone length and reduced but not eliminated visual function. However, the developmental phenotypes observed in mice do not match the same severity as the phenotypes observed in humans with Matthew-Wood Syndrome caused by mutant STRA6, suggesting that there might be alternate methods of retinol uptake by the RPE in mice though the majority of retinol uptake is accommodated by STRA6 normally [36]. This study was further supported by research conducted by the Noy lab the following year with a mouse STRA6 knockout where they found that retinoid homeostasis in tissues other than the eye was normal and that the mild loss in visual function from deletion of STRA6 in the mice is due to the high metabolic turnover of vitamin A in the eye without sufficient renewal by alternate retinol uptake methods by the RPE [37]. The von Lintig lab generated a novel STRA6 knockout mouse model to further establish the role of STRA6 in maintaining vitamin A homeostasis in ocular development and function, as well as gain a greater understanding of how STRA6 related diseases such as Matthew-Woods Syndrome are caused and treated. Their research in 2014 established STRA6 as the primary retinol transporter from the blood into the RPE and during development, and that vitamin A deficient mutant mice exhibited diseased phenotypes as previous studies, which were rescued to normal visual function by treatments of retinoid doses [38].

### 4.2. Retinol Binding Protein 4 Receptor 2 (RBPR2) in Whole-Body Vitamin A Homeostasis

While STRA6 is expressed in several different organs and tissues, such as the RPE in the eye, it is not expressed in all tissues (Figure 2 and Figure 3). The liver is the main organ involved in the storage of retinoids, however, STRA6 is not expressed in hepatic tissues. Thus, an alternative transport protein is likely expressed in tissues that do not contain STRA6. Discovered by Alapatt and colleagues in 2013, the Retinol Binding Protein 4 Receptor 2 (RBPR2) was found to be the high-affinity RBP4-binding transport protein responsible for the uptake of RBP4- bound retinol in the liver with a similar function as STRA6 in the RPE, however, the efflux capabilities [39]. Publications from our lab showed shown that Rbpr2 was also highly expressed in 11.5 hpf zebrafish embryos at the start of ocular development and in the intestines and pancreas of zebrafish larvae; however, mammalian expression-function patterns of RBPR2 still need further study [40]. The intestinal uptake and efflux of dietary retinol have been theorized to be facilitated by transport proteins alongside passive diffusion, but the specific transport proteins involved were never fully elucidated [41]. Recently, the work from our lab showed that RBPR2 might be involved in the influx (and possible efflux) of retinol in the intestines from the expression patterns of larval zebrafish, though more studies must be performed to establish this function of RBPR2 with mammalian models [40].

RBPR2 functions similarly to STRA6 and shares structural homology with STRA6 according to computer simulations run on the sequence of RBPR2 compared to the structure of STRA6. RBP4 is proposed to bind to residues S268, Y272, L273 of RBPR2, an amino acid-binding domain that is partially conserved between RBPR2 and STRA6 as well [42]. Even though the structure of RBPR2 was calculated in silico for comparison with STRA6, the crystal structure remains to be solved. Furthermore, despite the similar functionality and kinetics, the binding affinity and flux of RBPR2 with retinol bound to RBP are still unknown. The methods used by the Mancia laboratory to isolate STRA6 for cryo-EM have been published [43] and could provide a viable guideline in isolating similar membrane proteins such as RBPR2 for structural and functional analysis.

The work of Alapatt and colleagues after discovering RBPR2 have suggested that RBPR2 may be a major regulator of vitamin A homeostasis in the liver, among other tissues where the protein is expressed. Deficiencies in RBPR2 might play a role in the development of insulin-resistant phenotypes given the protein’s interaction with RBP4, in which an excess of holo-RBP4 is linked to insulin resistance and glucose intolerance [39]. Mutant and deficient RBPR2 has also been linked to symptoms seen in vitamin A deficiency (VAD), such as night blindness, microphthalmia, shortening of rods and cones, and retinal degeneration in zebrafish despite the protein not being expressed in the eye [40,42]. The incidence of VAD phenotypes in the eyes of RBPR2 mutants shows the importance of RBPR2 in eye development and in maintaining vitamin A homeostasis, though further study will need to be conducted using mammalian models.

## 5. All-*Trans* Retinoic Acid as a Transcription Factor

Circulatory all-*trans* Retinol, once taken up by peripheral cells via its specific membrane receptor (STRA6 or RBPR2), typically will convert it into one of two different retinoids: 11-*cis* retinal for visual function or all-*trans* retinoic acid as a transcription factor. One of the most important functions of all-*trans* retinoic acid is its contribution as a ligand in the RAR-RXR nuclear receptor. RAR and RXR constitute the two components of the dimer that serve to bind to retinoic acid response elements (RAREs) to activate the transcription of genes associated with the RARE. RAR-RXR signaling is modulated in its differentiation of effects through combinations of isoforms [44]. In the first level of modularization, both RAR and RXR are separated into α, β, and γ subunits, with further levels of differentiation to allow for many effects depending on isoform combinations [44]. RA-dependent signaling is of particular importance in development, with its effects ranging from the development of the neural plate, development of heart structure, development of eye structures, and development in various brain structures [45].

## 6. Vitamin A Metabolites—Retinoids (All-*Trans*-Retinoic Acid) in Development

As mentioned previously, the role of RA in development is related to its role as a ligand within RA signaling through RAR-RXR and RAREs. One of the premier examples of RA signaling is its role in the development of the brain. The proper localization of RA signaling is extremely important in the development of the hindbrain or rhombencephalon, as uncontrolled RA exposure is extremely teratogenic. This necessitates its layers of tight control during development [46]. One such layer of regulation involves the cytochrome p450, 26 (Cyp26a1) protein, which catalyzes the conversion of all-*trans* retinoic acid into relatively inactive 4-hydroxy-retinoic acid and 4-oxo-retinoic acid [47]. These conversions by Cyp26a1 serve to remove signaling active all-*trans* retinoic acid during specific stages of development; in the specific case of the hindbrain, allowing for a “gradient” of RA signaling for controlled induction of RA signaling related development [48,49]. Cyp26a1 and its redundant analogs generate a “stepwise” pattern of all-*trans* retinoic acid concentrations that allow for the proper pacing of development for both anterior and posterior sides. In Cyp26a1 knockout (*cyp26a1^−/−^*) zebrafish subjected to RA deficiency, a treatment of RA intended to rescue the fish caused the fish to exhibit teratogenic posteriorization of the hindbrain such as those found in wildtype fish exposed to 40 times that amount [46].

All-*trans* retinoic acid also has a major role in the proper development of the eye through its role in retinoic acid receptor (RAR) signaling. Given the ubiquitous nature of RA signaling in the multitude of different pathways in eye development and beyond, a full deserved explanation of the role of RA signaling in eye development is beyond the scope of this paper. One such role that RA signaling has is in the development of the lens. RA signaling is responsible for the formation of the lens placode through the combination of the optic vessel (OV) and prospective lens ectoderm (PLE). RA signaling is initially responsible for the activation of several genes that encode transcription factors, which include *Lhx2*, *Mab21l2*, *Rx*, and *Hes1*. These transcription factors promote the expression of the bone morphogenic protein 4 (BMP4) and bone morphogenic protein 7 (BMP7), which are both responsible for the activation of *Six3* and *Pax6* in the PLE, leading to the formation of the lens placode [50].

As mentioned in the above sections, mutations within the retinoid transport proteins of STRA6 and RBPR2 can lead to phenotypes associated with Matthew-Wood Syndrome, such as anophthalmia, microphthalmia, and night blindness [33,40,42]. The similarity in phenotypes for mutations in either of these transporters is likely explained by the unified theme of VAD, as deficiencies in these transporters should decrease the number of absorbed carotenoids. However, it has been shown that it is possible to rescue phenotypes expressing deficiency in these transporters. Zebrafish with a mutant RBPR2 containing a defective RBP4 binding domain (rbpr2^fs−muz99^); thereby disrupting the proper docking of RBP4 bound retinol to the transporter, causes rod and cone dystrophy, decreased ocular retinoid content, and a significant decrease in expression of enzymes involved in the processing and regulation of retinoids [42]. However, it was found that a solution of all-*trans* retinoic acid was able to fully rescue the mutant phenotypes, given that the treatment was given before the gastrulation stage and at a high enough concentration [42]. More interestingly, an injection of wild-type RBPR2 mRNA into mutant phenotypes at the 1-2 cell stage was able to fully rescue the mutant phenotypes [42]. 

VAD independent of mutations in retinoid transporters has been studied extensively by John Dowling, who is one of the pioneers in the role of vitamin A in eye development and a premier expert in the vertebrate retina as a whole. One such method to induce VAD is through chemical inhibition of retinaldehyde dehydrogenase (Raldh), the enzyme responsible for catalytically converting retinal into signaling active all-*trans* retinoic acid. One of the Raldh chemical inhibitors used in John Dowling’s work includes diethylaminobenzaldehyde (DEAB), which induced impaired development of the eye in zebrafish models [51]. In the DEAB experiment, treated zebrafish exhibited impaired retinal development in the 36, 60, 84 hpf larval stages and significant microphthalmia at the 5.5 dpf stage. Retinal histological sections of the eyes of zebrafish at the previously mentioned stages of development displayed a significant lack of characterization of retinal layers. Additionally, electroretinogram (ERG) measurements indicate that DEAB treated 5.5 dpf zebrafish exhibited null levels of electrical responses in response to a light stimulus when compared to non-treated zebrafish [51].

## 7. Vitamin A in Vision

Vitamin A is one of the key components involved with visual function. The main visual chromophore responsible for the vision that binds to opsin in photoreceptors is 11-*cis* retinal, a metabolite of vitamin A, which was discovered to be important for vision in the late 1960s by George Wald, who also elucidated the visual or retinoid cycle involved with the metabolism of vitamin A in visual function [52]. At the start of the visual cycle, all- trans-retinol enters the RPE through facilitated transport by STRA6 where it gets metabolized into a retinyl ester via the action of LRAT, which then converts to 11-*cis* retinol and subsequently 11-*cis* retinal through catalysis by RPE65 and retinol dehydrogenase 5 (RDH5), respectively [53]. 11-*cis* retinal is transported from the RPE into the photoreceptors by interphotoreceptor RBP (IRBP) and binds to opsin in either the rods or cones (forming rhodopsin or cone opsin, respectively) [52]. The holo-opsin complex becomes activated through a light catalyzed cis-trans isomerization of the opsin-bound 11-*cis* retinal into all-*trans* retinal-bound opsin, causing a photobleaching process where rhodopsin forms several different intermediate states that trigger a G-protein signaling pathway [54]. After photobleaching, all-*trans* retinal is hydrolyzed from opsin and converted back into all-*trans* retinol by RDH8 [55] before localizing back to the RPE by IRBP to repeat the visual cycle. An alternate cone visual cycle has recently been discovered in cone-dominant retinas of chickens and ground squirrels between cones and Müller cells where 11-*cis* retinol is regenerated in the Müller cells and transported back to the cones where 11-*cis* retinol is oxidized into 11-*cis* retinal for photoactivation of opsin [56]. This alternate cone visual cycle is believed to act in tandem with the classical visual cycle to maintain visual chromophore concentrations for cones in situations of bright light [57]. 

Dysfunction with enzymes involved throughout the visual cycles leads to several different retina pathologies. Leber Congenital Amaurosis (LCA) is caused by mutant RPE65 impacting the isomerization of all-trans retinol to 11-cis retinol, leading to childhood blindness [58,59]. Retinitis pigmentosa can be caused by dysfunctional LRAT, mutant RPE65, and P23H mutant opsin [52,58,59]. Retinal degeneration, among other pathologies caused by dysfunctional retinoid transport proteins, as noted earlier in this article [52]. Recently, a small molecule treatment was found to rescue the normal phenotype of opsins with P23H mutations in cell lines by the labs of Palczewski and Chen, with the treatment also providing protection from retinal degeneration in RDH8 and ABCA4 knockout mice showing its broad therapeutic use [60].

## 8. Concluding Remarks and Future Directions

To summarize this article, we discuss the overall transport of vitamin A and the enzymes involved from the intake of dietary vitamin A in the intestines to vitamin A storage in the liver and to the functional endpoints for vitamin A in the eye for visual function and as a transcription factor. Specifically, the transport proteins RBP4, STRA6, and RBPR2 were discussed in detail along with their known functions, structures, and pathologies caused by dysfunction or mutation of those proteins in various vertebrate models. Much is still needed to be uncovered regarding the storage, release, and transport of vitamin A (Figure 4). The crystal structure of RBPR2 has still not been solved as well as the functional understanding that comes from that. Furthermore, the possible efflux potential of RBPR2 for all-*trans* retinol, such as that seen in STRA6, from the intestines and liver has yet to be studied. Finally, it is still unknown how the eye signals to the liver (or peripheral tissues) to release vitamin A stores when retinoid concentrations are low. Despite how much is known of vitamin A, from its function to its transport, there is still much to study and discover with regards to this class of nutrients.

## Figures and Tables

**Figure 1 nutrients-13-03987-f001:**
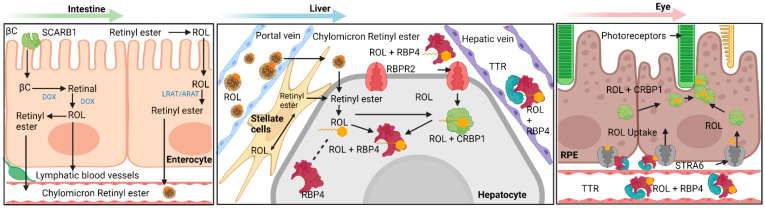
Overview diagram detailing the various pathways within vitamin A transport and metabolism into biologically useful forms. Generally, this process can be separated into four different categories: absorption, storage, release, and uptake. A succinct summary of these four categories is shown here for vitamin A: absorption of provitamin A into intestinal epithelial cells and initial processing into chylomicron bound retinyl esters, storage of retinyl esters within hepatic stellate cells, release of storage retinyl esters as RBP4 bound retinol into the bloodstream and uptake by the retinal pigment epithelium. Note the importance of STRA6 and RBPR2 as major facilitators in the transport of RBP4 bound retinol. βC—β-Carotene; SCARB1—Scavenger Receptor Class B, Type 1; LRAT—Lecithin Retinol Acyltransferase; ARAT—Acyl-CoA Retinol Acyltransferase (ARAT); ROL—All-*Trans* Retinol; STRA6 – Stimulated by Retinoic Acid 6; RBPR2—Retinol Binding Protein 4 Receptor 2; RBP4—Retinol Binding Protein 4; TTR—Transthyretin; CRBP1—Cellular Retinol Binding Protein 1; CRBP2—Cellular Retinol Binding Protein 2; RPE—Retinal Pigment Epithelium. Created with BioRender.com.

**Figure 2 nutrients-13-03987-f002:**
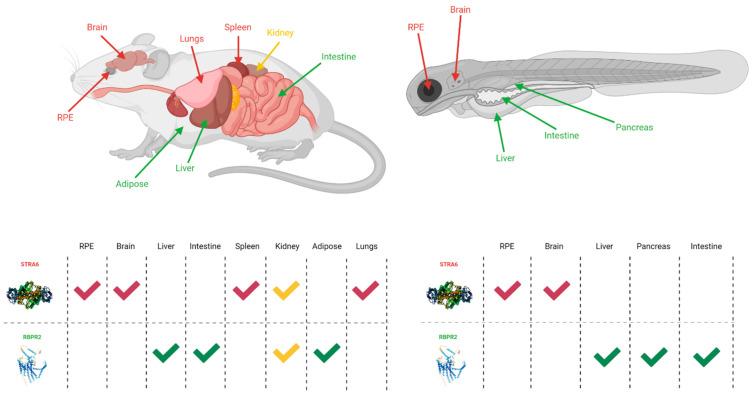
The expression sites of STRA6 and RBPR2 in mouse and zebrafish models [30,35,38,39,40]. Red arrows and checkmarks indicate STRA6 expression in that particular organ, green arrows and checkmarks indicate RBPR2 expression in that particular organ, and yellow arrows and checkmarks indicate expression of both STRA6 and RBPR2 in that organ. STRA6 is consistently expressed in the retinal pigment epithelium, brain, and shares expression with RBPR2 within the kidneys of both organisms. Otherwise, STRA6 and RBPR2 expression is mutually exclusive. Created with BioRender.com.

**Figure 3 nutrients-13-03987-f003:**
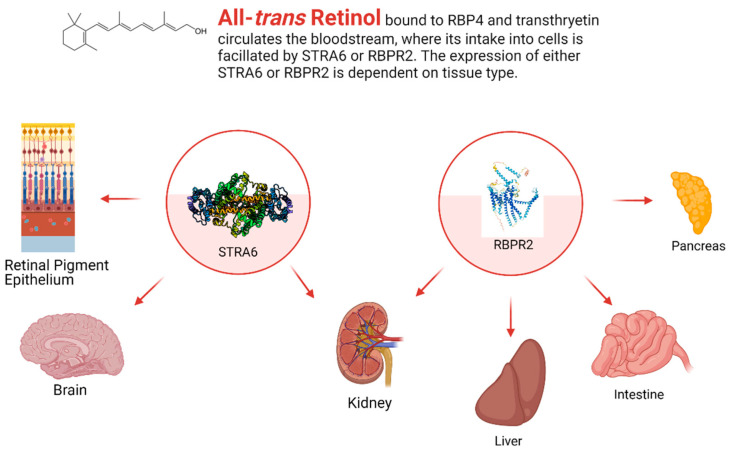
The general expression pattern of STRA6 and RBPR2. Again, STRA6 and RBPR2 share expression within the kidney, with mutually exclusive expression in all other organ systems [30,35,39,40]. Created with BioRender.com.

**Figure 4 nutrients-13-03987-f004:**
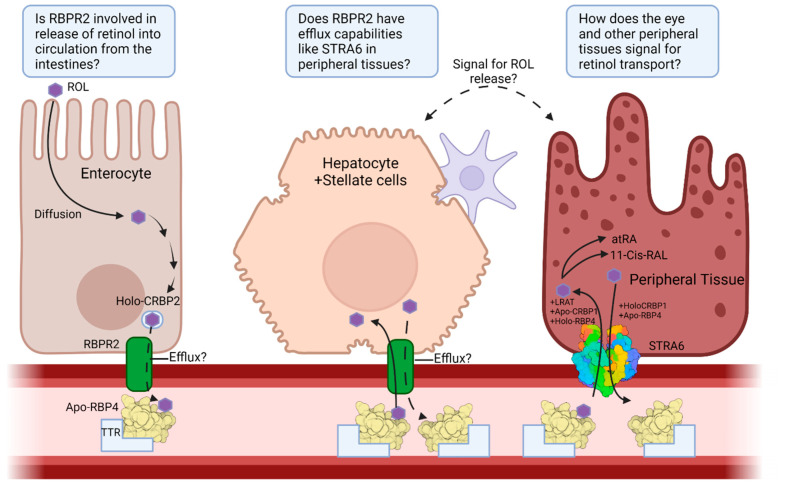
Overview of the transport pathway of vitamin A and the important proteins involved from its entrance through the enterocytes, to its influx and efflux from storage in hepatocytes, and its entrance to peripheral tissues. Questions about RBPR2 function in the intestines, probable efflux capability, and signaling mechanism for vitamin A release into serum are included. ROL—All-*trans* Retinol; CRBP1—Cellular Retinol Binding Protein 1; CRBP2—Cellular Retinol Binding Protein 2; STRA6—Stimulated by Retinoic Acid 6; RBPR2—Retinol Binding Protein 4 Receptor 2; RBP4—Retinol Binding Protein 4; TTR—Transthyretin; atRA—All-*Trans* Retinoic Acid; 11-Cis-RAL—11-Cis-Retinal; Apo—Unbound state; Holo—Bound state. Created with BioRender.com.

## Data Availability

Not applicable.

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
