# Peer review of "Vitamin A Transporters in Visual Function: A Mini Review on Membrane Receptors for Dietary Vitamin A Uptake, Storage, and Transport to the Eye"

_nutrients, 2021, doi:10.3390/nu13113987_

Round 1
Reviewer 1 Report
This is a generally well written review on vitamin A transport. There are a few items that need to be corrected or clarified:
Line 13, 225, 228, 232 for example: “Vitamin A” should not use the capital letter “V” for vitamin A in the middle of a sentence. This occurs in several locations.
Line 16: Direct intake of retinyl esters from animal source food is missing in the introduction. For all omnivores and carnivores, this is an important, if not the dominant source of vitamin A.
Line 26: Retina and liver should not be categorized together. Retina and RPE should be in the same category.
Line 156: RBP binds to STRA6 at nM affinity. Even without TTR, there is sufficient RBP4 in the blood to saturate STRA6. Therefore, it is not surprising that TTR knockout does not have severe phenotypes.
Even complete null of RBP4 does not cause systemic phenotypes in mice under standard lab diet, which contains high and excessive amount of vitamin A that promotes the retinyl ester pathway of delivery. Under vitamin A deficient conditions that mimic the natural environment, Quadro and colleagues showed that RBP4 knockout causes severe developmental defects that are consistent with the role of RBP in delivering vitamin A during development.
Line 221: Human and mouse STRA6 null do not match developmental phenotype, not adult vision phenotype because human STRA6 null all have no eyes (therefore it is impossible to study RPE and adult vision phenotype).
The vision phenotype of STRA6 knockout mice and lack of major defects in other organs consistent with its ligand RBP4’s knockout mice, which also have largely vision defects under standard lab diet. However, as mentioned above, Under vitamin A deficient conditions that mimic the natural environment, RBP4 knockout mice have severe developmental defects that match the severe developmental defects in STRA6 null humans.
Line 242: Liver does not need to express the RBP receptor to absorb vitamin A from holo-RBP because the liver is the primary site of vitamin A storage and the primary source of holo-RBP secretion. If the liver has abundant STRA6 expression, a short circuit would be formed that causes liver takes up the vitamin A it secretes in the form of holo-RBP. This short circuit would be wasteful and does not make biological sense. Therefore, it is expected that the liver does not express STRA6 or the RBP receptor to take up vitamin A from holo-RBP (which it secrets). It is well established that the human liver absorbs most vitamin A from food in the form of chylomicrons because vitamin A is absorbed through the small intestine in the same pathway as fat or other fat-soluble vitamins.
However, it is possible that there is a protein that exists in the liver to promotes holo-RBP formation and secretion (instead of absorption). RBPR2 may serve this function if it has the same activity as STRA6 in promoting retinol loading into apo-RBP (STRA6 is known to strongly promote the loading of retinol into apo-RBP, much more efficient than apo-RBP’s natural uptake of retinol).
Reviewer 2 Report
This manuscript by Ask et al. is a narrative review on the absorption, transport, uptake, and functions of vitamin A, emphasizing the cell membrane receptors for vitamin A (STRA6 and RBPR2) and the function of vitamin A in the eye. The manuscript is very well written and organized; the figures are vivid. This review is timely in terms of summarizing previous research and indicating future research directions. A few minor edits are needed.
1) In section 1 - intestinal absorption of vitamin A, the authors did not mention the process of chylomicron formation and metabolism, and the uptake of chylomicron remnants by liver, which is how the body processes newly ingested vitamin A. It should be added in this section.
2) Line 67, "esterification of storage retinyl esters into all-trans retinol". Did the authors want to say "hydrolysis of storage retinyl esters into..."?
3) Line 117, "almost all carotenoids", it's better to say "all retinoids"
4) Line 137-140, the authors first said all-trans retinol is "the fundamental transport form of vitamin A", then said retinyl esters "acts as the primary transport form of vitamin A". This can be very confusing. Retinyl esters acts as the primary transport form of newly ingested vitamin A. Revise to clarify.
5) Line 150-152, reference is needed.
6) Line 220, briefly define Matthew-Wood Syndrome.
7) Figure 2, add the names of the model organisms.
8) Figure 2, Stra6 was also found to be expressed in the lung of rodents.
9) Figure 3 is kind of a repeat of Figure 2 without providing much new information. Consider taking out Figure 3 or combine the two figures.
10) Line 343. "VAD in of itself independent of.." Reword this sentence.
